# Pioglitazone Ameliorates Acute Endotoxemia-Induced Acute on Chronic Renal Dysfunction in Cirrhotic Ascitic Rats

**DOI:** 10.3390/cells10113044

**Published:** 2021-11-05

**Authors:** Szu-Yu Liu, Chia-Chang Huang, Shiang-Fen Huang, Tsai-Ling Liao, Nai-Rong Kuo, Ying-Ying Yang, Tzu-Hao Li, Chih-Wei Liu, Ming-Chih Hou, Han-Chieh Lin

**Affiliations:** 1Department of Medical Education, Medical Innovation and Research Office (MIRO), Taipei Veterans General Hospital, Taipei 11217, Taiwan; maru2637@gmail.com (S.-Y.L.); cchuang7@vghtpe.gov.tw (C.-C.H.); knairong@gmail.com (N.-R.K.); 2Department of Medicine, Taipei Veterans General Hospital, Taipei 11217, Taiwan; sfhuang6@vghtpe.gov.tw (S.-F.H.); mchou@vghtpe.gov.tw (M.-C.H.); 3Faculty of Medicine, School of Medicine, National Yang-Ming Chiao Tung University, Taipei 11217, Taiwan; tlliao@vghtc.gov.tw (T.-L.L.); pearharry@yahoo.com.tw (T.-H.L.); cwliu2@vghtpe.gov.tw (C.-W.L.); 4Department of Medical Research, Taichung Veterans General Hospital, Taichung 11217, Taiwan; 5Division of Allergy, Immunology, and Rheumatology, Department of Internal Medicine, Shin Kong Wu Ho-Su Memorial Foundation, Taipei 11217, Taiwan

**Keywords:** endotoxemia, lipopolysaccharide, pioglitazone, cirrhosis, PPARγ, TNFα

## Abstract

Endotoxemia-activated tumor necrosis factor (TNFα)/nuclear factor kappa B (NFκB) signals result in acute on chronic inflammation-driven renal dysfunction in advanced cirrhosis. Systemic activation of peroxisome proliferator-activated receptor gamma (PPARγ) with pioglitazone can suppress inflammation-related splanchnic and pulmonary dysfunction in cirrhosis. This study explored the mechanism and effects of pioglitazone treatment on the abovementioned renal dysfunction in cirrhotic rats. Cirrhotic ascitic rats were induced with renal dysfunction by bile duct ligation (BDL). Then, 2 weeks of pioglitazone treatment (Pio, PPAR gamma agonist, 12 mg/kg/day, using the azert osmotic pump) was administered from the 6th week after BDL. Additionally, acute lipopolysaccharide (LPS, Escherichia coli 0111:B4; Sigma, 0.1 mg/kg b.w, i.p. dissolved in NaCl 0.9%) was used to induce acute renal dysfunction. Subsequently, various circulating, renal arterial and renal tissue pathogenic markers were measured. Cirrhotic BDL rats are characterized by decreased mean arterial pressure, increased cardiac output and portal venous pressure, reduced renal arterial blood flow (RABF), increased renal vascular resistance (RVR), increased relative renal weight/hydroxyproline, downregulated renal PPARγ expression, upregulated renal inflammatory markers (TNFα, NFκB, IL-6, MCP-1), increased adhesion molecules (VCAM-1 and ICAM-1), increased renal macrophages (M1, CD68), and progressive renal dysfunction (increasing serum and urinary levels of renal injury markers (lipocalin-2 and IL-18)). In particular, acute LPS administration induces acute on chronic renal dysfunction (increasing serum BUN/creatinine, increasing RVR and decreasing RABF) by increased TNFα-NFκB-mediated renal inflammatory markers as well as renal M1 macrophage infiltration. In comparison with the BDL+LPS group, chronic pioglitazone pre-treatment prevented LPS-induced renal pathogenic changes in the BDL-Pio+LPS group. Activation of systemic, renal vessel and renal tissue levels of PPARγ by chronic pioglitazone treatment has beneficial effects on the endotoxemia-related TNFα/NFκB-mediated acute and chronic renal inflammation in cirrhosis. This study revealed that normalization of renal and renal arterial levels of PPARγ effectively prevented LPS-induced acute and chronic renal dysfunction in cirrhotic ascitic rats.

## 1. Introduction 

In cirrhosis, endotoxemia-activated tumor necrosis factor α (TNFα)/NFκB signals are involved in the development of systemic, pulmonary, splanchnic and renal dysfunction [1,2,3]. In cirrhotic kidneys, acute accumulation of lipopolysaccharide (LPS, endotoxin) results in TNFα/NFκB-mediated hypoperfusion and inflammation-driven acute on chronic renal dysfunction by increasing renal adhesion molecule, increasing renal M1 macrophage infiltration, decreasing renal blood flow (RABF), increasing renal vascular resistance (RVR), and increasing renal tissue/renal vascular inflammation [4,5,6,7,8].

Endotoxemia-related renal inflammation is characterized by a reduction in RABF in septic rats [8]. Endotoxin and TNFα/NFκB are involved in the RABF decline and chronic renal dysfunction in bile-duct ligated rats [9,10]. In healthy rats, the infusion of TNFα lowers RABF and increases RVR by inducing vascular inflammation without affecting arterial pressure [11,12]. In cirrhotic ascitic patients, decreasing the levels of circulating endotoxin and TNFα using selective intestinal decontamination with rifaximin improved chronic renal dysfunction [13]. During renal vascular inflammation, a significant (around 40%) reduction in RABF and chronic renal dysfunction were reported in compensated cirrhotic patients compared to those in healthy volunteers [1]. PPARγ is expressed in renal medullary interstitial cells in the juxtaglomerular apparatus and glomeruli, including podocytes, mesangial cells, and renal microvascular endothelial cells [14]. Given that multiple renal cell types have endogenous PPARγ expression and activity, its activation in the kidney may be critical for governing renal function. In mice with diabetic nephropathy, chronic pioglitazone treatment decreases both renal vascular inflammation and reduces RVR [15].

Decreased pulmonary expression of PPARγ accelerated the ongoing endotoxemia-related TNFα/NFκB-mediated lung inflammation and injury [16]. In septic rats, pharmacological activation of PPARγ attenuates endotoxin-induced TNFα/NFκB-mediated renal injury and dysfunction [17,18]. Activation of PPARγ with pioglitazone reduces renal macrophage infiltration [19,20].

Activation of endothelial PPARγ with PPARγ agonist rosiglitazone inhibits LPS-induced vascular inflammation [21]. Pioglitazone is a nuclear receptor PPARγ activator that exerts anti-inflammatory effects by antagonizing LPS-mediated vascular inflammation [22]. Downregulated hepatic PPARγ expression is associated with increased systemic circulating inflammatory cytokines in BDL rats [23]. In BDL-cirrhotic rats, chronic pioglitazone treatment attenuates motor and cognition impairments and suppresses TNFα/NFκB-mediated inflammation-related portosystemic shunting and hepatopulmonary syndrome [24,25,26].

Taken together, the effects of chronic pioglitazone treatment on endotoxemia-induced cirrhosis-related acute on chronic vascular and tissue inflammation-related renal dysfunction have not been explored. This study evaluated the mechanism and effects of activation of systemic, renal tissue and renal vascular levels of PPARγ by chronic pioglitazone treatment on the TNFα/NFκB-mediated acute on chronic renal dysfunction in cirrhotic ascitic rats.

## 2. Materials and Methods

### 2.1. Methods

Common bile duct ligation (BDL) was conducted on adult male Sprague-Dawley rats (300–350 g), as described previously [2,5,10,24]. All animal experiments were approved by the Animal Care Committee of the National Yang-Ming Chiao Tung University (YMCU) and conducted in the animal facilities of YMCU with No. 1090211r which was approved on 1 January 2020. All efforts were made to minimize the number of animals necessary to produce reliable results, and suffering was reduced by administering anesthetics (zoletil and xylocaine). At the end of the experiments, the rats were euthanized using a 2–3 times high anesthetic dose of zoletil.

To evaluate the effects of chronic pioglitazone (Pio, PPAR gamma agonist, 12 mg/kg/day, using the azert osmotic pump for intraperitoneal administration) or DMSO (a substance used to dissolve pioglitazone), pre-treatment on LPS was used to induce acute on chronic renal injury of cirrhotic rats. Four weeks after BDL, cirrhotic rats were randomized to receive two weeks of pioglitazone before the acute LPS challenge. Without modifying blood sugar levels, this dose of chronic pioglitazone treatment can ameliorate splanchnic inflammation, decrease portosystemic shunting, and prevent hepatopulmonary syndrome in cirrhotic animals [24,25,26]. Then, to induce an acute renal dysfunction, rats were randomly allocated 6 weeks after BDL to receive an intraperitoneal injection of LPS (*Escherichia coli* 0111:B4; Sigma, 0.1 mg/kg b.w, i.p. dissolved in NaCl 0.9%). Subsequently, various parameters were measured 3 h after LPS administration. The experimental groups were sham (*n* = 4), sham+LPS (*n* = 4), sham-Pio+LPS (*n* = 4), BDL (*n* = 9), BDL+LPS (*n* = 9), and BDL-Pio+LPS (*n* = 9) rats.

### 2.2. Urine Sample Collection

In order to evaluate the effects of chronic pioglitazone treatment on progressive cirrhosis-related renal injury, urinary renal tubular epithelial damage markers (uLipocalin-2, uIL-18 and creatinine (Colorimetric kits purchased by Cayman Chemical)) were measured in urine collected daily at 1–2, 15–16, 29–30, and 43–44 days after BDL. All the measurements and acute LPS infusion were conducted after the last daily urine collection. To collect urine, the rats were first caged in 24 h metabolic cages for 2 days of acclimatization to reduce separation effects.

### 2.3. Hemodynamic Measurements in the Days of Tissue Collections

Tissue and blood samples were collected after various hemodynamic measurements (mean arterial pressure (MAP), cardiac output (CO), heart rate (HR), bilateral RABF (mL/min·100 g body weight, BW), and portal venous pressure (PVP)). The right renal artery was identified at its aortic origin, and a 5 mm segment was gently dissected from the surrounding tissues. A pulsed-Doppler flow transducer (T206 small animal blood flowmeter; Transonic Systems, Ithaca, NY, USA) was then placed to measure the renal artery blood flow rate. Following a 1 h equilibrium period, the RABF was measured for 1 h. The results are reported as mL/min. Extrarenal renal vascular resistance (RVR) was calculated as MAP/RABF. The cardiac index (CI) was calculated using the following formula: CI=CO/BW. Stroke volume (SV; mL/beats) was calculated as (CO (mL/min)/HR (beats/min))).

### 2.4. Measurement of Various Plasma Pathogenic Factors

Blood was obtained from the inferior vena cava at the time of euthanasia. Plasma levels of biochemical parameters were determined using an automated biochemistry analyzer (Olympus, Tokyo, Japan). Additionally, serum samples were analyzed for TNF-α and IL-6 levels by enzyme-linked immunosorbent assay according to the manufacturer’s instructions (BioSource International, Camarillo, CA, USA).

### 2.5. Isolated Renal Perfusion Study

All rats were anesthetized intraperitoneally with zoletil (50 mg/kg body weight, ip) and fixed in the supine position. The isolated renal perfusion study of the right kidney was conducted as previously described [10,12]. In the rat perfusion system, RVR was recorded using a pressure transducer (Gould, Oxnard, CA, USA) as changes in renal perfusion pressure (RPP) downstream from the pump. RVR (mm Hg/mL per min/g) was calculated from the ratio of constant perfusion flow to the RPP. Once the RPP reached its steady state, experiments were initiated by the addition of cumulative concentrations of TNFα (0.1, 0.3, and 0.5 ng/g/min) to the perfusion apparatus with Krebs–Henseleit solution inside. Different concentrations of TNFα were added after the previous response reached a maximum.

### 2.6. Tissue Profiles

All renal arteries (including renal, lobar, and arcuate arteries) and kidneys were collected, immediately frozen in liquid nitrogen, and stored at −80 °C until analysis. In addition to immunochemistry and immunofluorescence staining, periodic acid–Schiff (PAS)-stained and Sirius Red-stained renal sections were also prepared to evaluate the severity of renal tubular damage and tubulointerstitial fibrosis.

### 2.7. Flow Cytometry

For measuring macrophage infiltration, the cell pellets of renal tissue were washed in FACS buffer for staining with F4/80-FITC, CD11c-PE, and CD206-AF488 antibodies (BD Biosciences, Franklin Lakes, NJ, USA) and incubated on ice for 1 h. After the wells were washed in FACS buffer, they were re-suspended in 500 μL of FACS and analyzed with a FACS Calibur flow cytometer; the resulting data were analyzed using the FlowJo software (Tree Star, Ashland, OR, USA). Fluorescence voltages were determined using matched unstained cells. Two hundred thousand events were acquired in a live mononuclear gate. Then, the number of M1 (F4/80(+)/CD11c(+)) and M2 (F4/80(+)/CD206(+)) macrophages in 1 mL of tissue homogenates was obtained.

### 2.8. Materials

Antibodies against TNFα, IL-6, CD68, CD163, MCP-1, F4/80-FITC, CD11c-PE, and CD206-AF488 were purchased from Cell Signaling Technology (Danvers, MA, USA) and Santa Cruz Biotechnology (Santa Cruz, CA, USA). Primers (Table 1) of TNFα, MCP-1, IL-4, IL-13 and 18S were purchased from Applied Biosystems. All other reagents were obtained from Sigma (St. Louis, MO, USA). Tissue levels of the adhesion molecules ICAM-1, VCAM-1 and MCP-1 were measured by enzyme-linked immunosorbent assay (ELISA) using commercially available ELISA kits (BD Bioscience, San Jose, CA, USA).

### 2.9. Statistical Analysis

All values are expressed as the mean ± standard error of the mean (SEM). Differences between groups were compared using Mann–Whitney U test for the comparison of the data of mean/SEM, and differences between two groups and ANOVA with post-hoc test for comparison among multiple groups. Statistical significance was set at *p* < 0.05.

## 3. Results

### 3.1. Cirrhotic BDL Rats Are Characterized by Progressive Renal Dysfunction that Can Be Attenuated by Chronic Pioglitazone Treatment

In comparison with sham rats, cirrhotic rats were characterized by decreased MAP, increased CO and PVP, reduced RABF, increased RVR, increased relative renal weight and increased renal hydroxyproline levels (Table 2). Although not reaching significance, acute LPS administration showed a decreasing MAP and CO and increasing PVP and RABF. Acute on chronic renal dysfunction (increased blood urea nitrogen and creatinine, Figure 1C,D) were observed in BDL+LPS groups. Notably, the relative kidney weight and renal hydroxyproline were not different between the sham, sham+LPS, BDL, and BDL+LPS groups. In comparison with the sham group, a serial increase in renal injury markers (IL-18 and lipocalin-2) in urine and renal tissue were observed (Figure 1A,B and Figure 2A) in BDL groups. This observation indicated that progressive injury-related renal dysfunction existed in rat cirrhotic kidneys two weeks after BDL. In the BDL-Pio+LPS group, before LPS administration, the urinary levels of IL-18 and lipocalin-2 were lower than those in the BDL group (Figure 1A,B). These results indicated that the two weeks of pioglitazone treatment attenuated progressive renal dysfunction as well as acute LPS-deteriorated renal function in cirrhotic rats. Additionally, this pre-treatment attenuated the LPS-induced decrease in MAP and CO and the increase in RABF, serum BUN, and serum creatinine in the BDL-Pio+LPS group (Table 2 and Figure 1C–D).

### 3.2. Chronic Pioglitazone Treatment Suppressed Serum Endotoxin, TNFα, IL-6, ALT and Total Bilirubin in Advanced Cirrhotic Rats

Notably, cirrhotic rats were characterized by higher circulating TNFα, IL-6, VCAM-1, ICMA-1, ALT, total bilirubin (TB) and lower serum albumin than rats in the sham group (Table 3 and Figure 1E,F). Acute LPS administration significantly increased circulating TNFα, IL-6, VCAM-1, ICAM-1, ALT, TB and decreased serum albumin levels in the BDL+LPS group. In particular, the chronic pioglitazone pre-treatment prevented the LPS-induced increase in serum TNFα, IL-6, VCAM-1, ICAM-1, ALT, TB and decreased serum albumin in BDL-Pio+LPS rats. The fasting blood sugar (FBS) level was slightly higher in the BDL group than in the sham group. Nonetheless, FBS was not affected by acute LPS administration and chronic pioglitazone treatment in sham-LPS, sham-Pio+LPS, BDL-LPS, and BDL-Pio+LPS rats.

### 3.3. Acute LPS Administration Downregulated Renal PPARγ Expression and Increased Renal M1 Macrophage Infiltration and Inflammation in Cirrhotic Ascitic Rats

In comparison with the sham group, the downregulation of renal PPARγ expression was accompanied by the upregulation of renal TNFα, NFκBp65, IL-6 and increased renal macrophage infiltration (upregulated macrophage marker CD68) in the BDL group (Figure 1H,I and Figure 2B,C). In the BDL group, the frequency of renal M1 macrophage infiltration (increased levels of TNFα and MCP-1 in cell lysates) was higher than that in the sham group (Figure 1H, Figure 3A–D and Figure 4A). The decrease in the percentage of renal M2 macrophages was companied by decreased levels of M2 marker (IL-4 and IL-13) in cell lysates of renal tissue of the BDL group (Figure 2B and Figure 3C,D). Particularly, acute LPS administration induced a further increase in renal M1 macrophage infiltration, suppression of renal PPARγ, and upregulation of renal TNFα, NFκBp65, IL-6 and MCP-1. Chronic pioglitazone pre-treatment attenuated the above-mentioned LPS-related infiltrated macrophage-mediated pathogenic changes in the BDL group (Figure 1, Figure 2 and Figure 3).

### 3.4. Effects of Chronic Pioglitazone Pre-Treatment Suppressed LPS-Induced TNFα -Mediated Renal Injury and Fibrosis in Cirrhotic Ascitic Rats

Pioglitazone pre-treatment attenuated LPS-induced renal tubular injury (IL-18, Figure 2A), inflammation (TNFα, IL-6, MCP-1, Figure 1G–I, Figure 2B, Figure 3B and Figure 4A), tubulointerstitial injury and fibrosis (PAS-stained and Sirius Red-stained data, Figure 2C and Figure 3A) by activating renal PPARγ expression. In addition, pioglitazone reduced the mortality rate of BDL rats to 32.9% during the 3 h following LPS injection compared with the saline treated group (*p* < 0.05).

### 3.5. Chronic PPARγ Agonist Pioglitazone Pre-Treatment Attenuates the LPS-Induced TNFα-Mediated Increase in Renal Vascular Resistance (RVR) in Cirrhotic Ascitic Rats

Figure 4B,C shows that the cumulative concentrations of TNFα induced an increase in RVR and a decrease in RABF in sham-perfused kidneys. Significantly, the degrees of TNFα-induced increase in RVR and decrease in RABF were higher in BDL-perfused rat kidneys than those in sham-perfused kidney. Furthermore, acute LPS administration significantly increased the magnitude of the TNFα-induced increase in RVR and the decrease in RABF both in sham+LPS and BDL+LPS groups, whereas the degree of changes was higher in the BDL+LPS group than those in the sham+LPS group. In BDL rats with chronic pioglitazone pre-treatment, a lower degree of LPS-enhanced TNFα-induced increase in RVR and decrease in RABF were noted in the BDL-Pio+LPS group than in the BDL+LPS group (Figure 4B,C).

### 3.6. Chronic Pioglitazone Pre-Treatment Attenuated LPS-Induced TNFα/NFκB-Mediated Renal Tissue and Renal Vascular Inflammation in BDL Rats

In comparison with the sham group, lower PPARγ expression was associated with increased levels of inflammatory mediators (TNFα, IL-6, MCP-1, NFκBp65, and CD68, Figure 5A,B,E,H) in the renal arterial tissue of the BDL group. Furthermore, acute LPS pre-administration significantly downregulated PPARγ expression, increased M1 macrophage infiltration, and increased vascular inflammation in renal arteries of the BDL+LPS group. In particular, chronic pioglitazone pre-treatment attenuated the LPS-induced TNFα/NFκB-mediated pathogenic changes in the renal arterial tissue of the BDL-Pio+LPS group.

## 4. Discussion

In this study, cirrhotic portal hypertensive rats were characterized by reduced RABF, increased RVR, upregulated renal inflammatory/adhesion molecules markers, and progressive renal dysfunction. In particular, acute endotoxin (LPS) administration induced acute on chronic renal dysfunction by increasing TNFα-NFκB-mediated renal inflammatory markers. Chronic pioglitazone pre-treatment prevented LPS-induced renal pathogenic changes in the cirrhotic group. Activation of systemic, renal tissue and renal vessel levels of PPARγ by chronic pioglitazone treatment has beneficial effects on the endotoxemia-related TNFα/NFκB-mediated acute on chronic renal inflammation in cirrhosis. This study revealed that normalization of renal and renal arterial levels of PPARγ effectively prevented LPS-induced acute and chronic renal dysfunction in cirrhotic ascitic rats.

Cirrhotic mice are suspected to develop LPS-induced TNFα-mediated AKI [27]. In cirrhotic patients with spontaneous bacterial peritonitis (SBP), those with renal impairment had significantly higher plasma and ascitic fluid TNFα levels than those without renal dysfunction [28]. In this study, the chronic inhibition of the levels of circulating, renal tissue and renal arterial TNFα by the chronic PPARγ agonist pioglitazone significantly improved renal function of cirrhotic ascitic rats (Figure 6).

TNFα directly induces lipocalin-2 and IL-18 production from renal epithelial cells, which are markers that represent the severity of renal injury [29,30,31,32]. In the Child–Pugh class of decompensated cirrhosis, urinary lipocalin-2/IL-18 levels increased and GFR decreased significantly [30,31,32]. The most common cause for acute on chronic renal dysfunction in cirrhosis is acute tubular necrosis (ATN), which occurs as a complication of sepsis [4,5,9,15,32]. Urine levels of IL-18 and lipocalin-2 from patients with cirrhosis discriminate between those with ATN and other types of kidney impairments [32]. Activation of renal PPAR-γ with pioglitazone suppressed renal IL-18 and lipocalin-2 expression in a renal ischemia-reperfusion model [33]. In our study, restoration of renal tissue PPARγ levels by chronic pioglitazone treatment suppressed the progressive TNFα-mediated increased RVR and renal injury (inhibition of urinary lipocalin-2 and IL-18) in cirrhotic ascitic rats with renal dysfunction. As shown in Table 2, in comparison with the BDL group, the acute intraperitoneal (IP) infusion of LPS induced a mean 195% (56.9 ± 8.1 vs. 168.3 ± 9.6 pg/mL) increase in serum TNFα level in the BDL+LPS group, whereas LPS only induced a 12% (56.9 ± 8.1 vs. 64.1 ± 7.1 pg/mL) increase in serum TNFα level in the BDL-pio+LPS group. This result indicated that pioglitazone treatment prevented 183% of the LPS-induced elevation of serum TNFα level in cirrhotic rats. Additionally, in comparison with liver injury markers (serum ALT level) of the BDL group, the 259% of elevation of liver injury markers in the BDL+LPS group was suppressed to 71%, which indicated that the pioglitazone treatment prevented 186% of the LPS-induced elevation of serum ALT (Table 2). Similarly, the pioglitazone treatment suppressed 40% of the LPS-elevated renal injury marker (renal IL-18 expression) in the BDL-pio+LPS group. Particularly, a higher degree of pioglitazone-suppressed liver injury marker (serum ALT) than renal marker (IL-18) might be the result of the direct access of LPS to the portal system by IP administration. These results indicated that the circulating TNFα mediated the LPS-induced liver and renal injury, and these changes in cirrhotic rats in this study can be ameliorated by pioglitazone treatment.

NFκB plays a major role in the inflammatory response, and over-activation of NFκB induces the overexpression of TNFα, thus accelerating renal injury [4,7,8,13,34]. In streptozotocin-induced diabetic nephropathy rats, chronic pioglitazone treatment reduced renal NFκB, IL-1β and IL-18 levels, depressed the glomerular mesangial expansion, and decreased serum BUN/creatinine [35].

Intercellular adhesion molecule-1 (ICAM-1) and vascular cell adhesion molecule-1 (VCAM-1) expressed on endothelial cells are involved in the interaction between leukocytes and endothelial cells [6,36]. Suppression of NFκB signaling inhibits TNFα-stimulated expression of ICAM-1 and VCAM-1 and the adhesion of monocytes to the human bronchial epithelial cell line [36]. In cholestasis, TNFα mediates IL-6 release from macrophages to aggravate renal dysfunction [7,9,10]. In a sepsis model, pioglitazone reduced LPS-induced TNFα and IL-6 production from mouse macrophages through inhibition of NFκB [37,38]. Pioglitazone was found to inhibit TNFα-induced expression of ICAM-1 and VCAM-1 in activated cultured endothelial cells and an ischemia/reperfusion renal injury model [33,39]. Notably, in cirrhotic ascitic rats from our study, chronic pioglitazone pre-treatment attenuated LPS-induced TNFα/NFκB-mediated acute on chronic renal dysfunction by suppressing renal IL-6, ICAM-1 and VCAM-1.

LPS can induce NFkB-mediated MCP-1 production in rat macrophages and renal tubular epithelial cells [40,41]. MCP-1 can stimulate glomerular macrophage infiltration and renal inflammation [42,43]. Increased renal macrophage infiltration is associated with progressive tubulointerstitial renal fibrosis in mice three weeks after BDL [44]. Cirrhotic patients with higher urine MCP-1 level have a higher probability of developing acute renal dysfunction [45]. Chronic pioglitazone protects patients from diabetic nephropathy by reducing urinary MCP-1 excretion and proteinuria [46]. In our current study, pioglitazone pre-treatment prevented LPS-induced acute on chronic renal dysfunction by inhibiting MCP-1-mediated renal macrophage infiltration and renal inflammation in cirrhotic ascitic rats.

M1 macrophages exert a pathogenic function in renal inflammation, whereas M2 macrophages appear to suppress inflammation and promote injury repair [47]. Increased M1 macrophage infiltration is a critical pathogenic factor for the initiation of LPS-induced or inflammation-driven renal dysfunction [48,49]. Activation of PPARγ with pioglitazone suppresses M1 macrophage polarization and skews circulating monocytes toward an anti-inflammatory M2 macrophage phenotype [19,20]. The CD68 molecule, which is highly expressed on tissue macrophages, is functionally important for M1 macrophages. Treatment with pioglitazone reduces CD68^+^ macrophage infiltration and MCP-1 release in adipose tissue [50]. In summary, chronic pioglitazone pre-treatment in cirrhotic ascitic rats effectively decreased LPS-induced M1 polarization of macrophages and renal dysfunction.

It has been reported that intraperitoneal (IP) administration of drugs in experimental animals is a justifiable route for pharmacological and proof-of-concept studies where the goal is to evaluate the effect(s) of target engagement rather than the properties of a drug formulation and/or its pharmacokinetics for clinical translation. A previous study had reported that the bioavailability and absorption for the IP route of small molecular agents (MW < 5000), such as pioglitazone (MW 392.9), are higher than those by oral route. However, both IP and oral routes have a similar degree of first pass metabolism of these small molecular agents in the liver [51]. In comparison with the oral route, the IP technique is easy to master and minimally stressful for animals. The IP route is especially commonly used in chronic studies involving rats for which repetitive oral access is challenging. In this study, two weeks of pioglitazone was administered by IP with an azert osmotic pump. Pioglitazone is well absorbed, has an oral bioavailability of about 80%, and is extensively metabolized to active and inactive metabolites in the liver [52,53,54,55]. In future studies, the effectiveness of oral administration of two weeks of pioglitazone is needed to be compared with the IP administration in this study.

A high prevalence of renal dysfunction has been reported among non-alcoholic steatohepatitis (NASH) patients [56]. Severe NASH is the most rapidly growing indication for simultaneous liver-kidney transplantation, with poor renal outcomes [57]. Several large-scale randomized controlled trials have reported the effectiveness of pioglitazone in treating NASH to improve markers of hepatic steatosis and fibrosis on liver histology [52,58]. A recent study reported that a low dose of pioglitazone was safe and effective in diabetes patients with CKD [53]. In particular, pioglitazone decreases the incidence of new-onset end-stage renal disease in diabetes patients [59]. Renal dysfunction in NASH and diabetic patients share a common pathogenesis [60,61]. Taken together, pioglitazone might have the potential to protect NASH patients from the development of renal dysfunction; this needs to be evaluated in future studies.

## 5. Conclusions

In conclusion, as shown in Figure 6, restoration of systemic, renal tissue and renal arterial PPARγ by chronic pioglitazone treatment attenuated cirrhosis-related renal dysfunction and endotoxemia-induced acute on chronic renal dysfunction.

## Figures and Tables

**Figure 1 cells-10-03044-f001:**
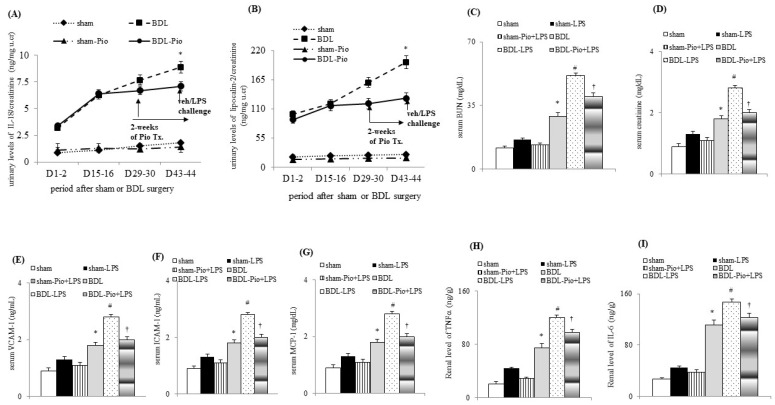
Chronic piogliotazone treatment attenuated chronic renal dysfunction in cirrhotic rats. (**A**,**B**) Urinary levels of renal injury markers including interleukin-18 (IL-18) and lipocalin-2; (**C**,**D**) serum BUN and creatinine; serum levels of (**E**) vascular cell adhesion molecule-1 (VCAM-1), (**F**) intracellular adhesion molecule-1 (ICAM-1), (**G**) monocyte chemoattractant protein-1 (MCP-1); renal levels of (**H**) TNFα, tumor necrosis factor α (TNFα) and (**I**) interleukin-6 (IL-6); * *p* < 0.05 sham vs. BDL group; # *p* < 0.05 vs. BDL vs. BDL-LPS group; † *p* < 0.05 vs. BDL-pio+LPS vs. BDL-LPS group.

**Figure 2 cells-10-03044-f002:**
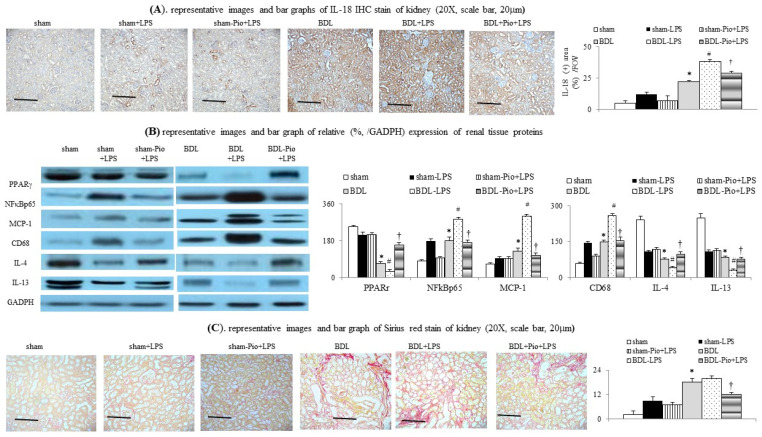
Chronic pioglitazone treatment suppressed LPS-induced acute on chronic renal dysfunction by prevention of increased renal macrophage infiltration. (**A**) Renal interleukin-18 (IL-18) expression; (**B**) various renal protein expressions between groups; (**C**) renal Sirius Red staining expression; * *p* < 0.05 vs. sham group; # *p* < 0.05 vs. BDL group; † *p* < 0.05 vs. BDL-LPS.

**Figure 3 cells-10-03044-f003:**
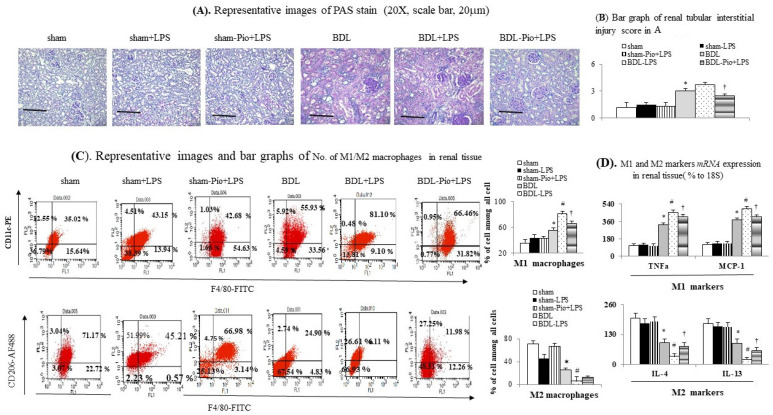
Chronic pioglitazone treatment suppressed the lipopolysaccharide (LPS)-induced acute on chronic renal injury by the prevention of increased renal M1 macrophages. (**A**,**B**) Renal PAS stain; (**C**) frequency of M1/M2 macrophages in renal tissue; (**D**). mRNA of M1/M2 markers in homogenates of renal tissue. * *p* < 0.05 vs. sham group; # *p* < 0.05 vs. BDL group; † *p* < 0.05 vs. BDL-LPS.

**Figure 4 cells-10-03044-f004:**
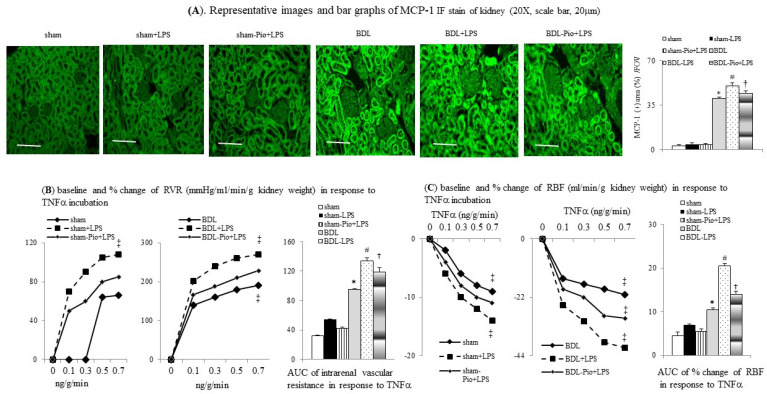
Chronic pioglitazone treatment inhibited the lipopolysaccharide (LPS)-enhanced TNFα and tumor necrosis factor α (TNFα)-induced increase in renal vascular resistance (RVR) of BDL-cirrhotic rats. (**A**) Immunofluorescence (IF) stain of monocyte chemoattractant protein-1 (MCP-1) expression in rat kidney. Concentration-response curve and bar graphs of AUC of (**B**) RVR and (**C**) renal blood flow (RABF) in response to cumulative concentrations of TNFα; * *p* < 0.05 vs. sham group; # *p* < 0.05 vs. BDL group; † *p* < 0.05 vs. BDL-LPS; ‡ *p* < 0.05 vs. lower concentration of TNFα.

**Figure 5 cells-10-03044-f005:**
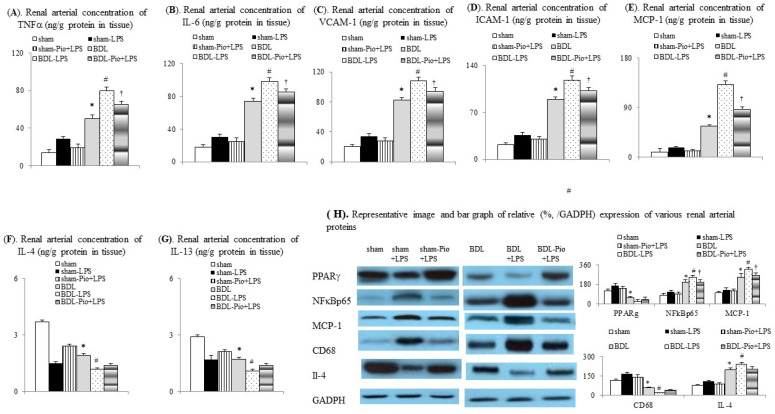
Chronic pioglitazone treatment inhibited the lipopolysaccharide (LPS)-induced vascular inflammation in rat cirrhotic renal arteries. Renal arterial concentrations of (**A**) TNFα, tumor necrosis factor α (TNFα), (**B**) interleukin-6 (IL-6), (**C**) vascular cell adhesion molecule-1(VCAM-1), (**D**) intracellular adhesion molecule-1 (ICAM-1), (**E**) monocyte chemoattractant protein-1 (MCP-1), (**F**) interleukin-4 (IL-4), and (**G**) interleukin-13 (IL-13). (**H**) Expression of various proteins in renal arteries. * *p* < 0.05 vs. sham group; # *p* < 0.05 vs. BDL group; † *p* < 0.05 vs. BDL-LPS.

**Figure 6 cells-10-03044-f006:**
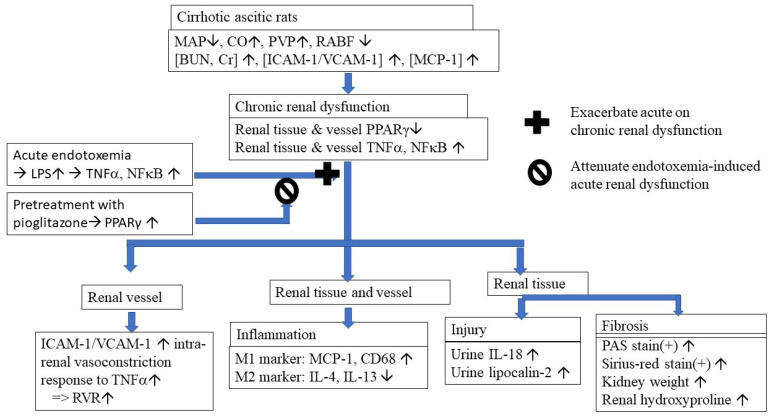
Graphical summary of the pathogenic mechanisms and effects of restoration of renal tissue and vessels PPAR with pioglitazone on the endotoxemia-induced acute on chronic renal dysfunction in cirrhotic rats. MAP: mean arterial pressure; CO: cardiac output; PVP: portal venous pressure; RABF: renal arterial blood flow; ICAM-1: intercellular adhesion molecule 1; VCAM-1: vascular cell adhesion molecule 1; MCP-1: monocyte chemoattractant protein 1; PPARγ: peroxisome proliferator-activated receptor gamma; TNFα: tumor necrosis factor alpha; NFkB: nuclear factor kappa-light-chain-enhancer of activated B cells; LPS: lipopolysaccharide; RVR: renal vascular resistance; M1/M2: two types of macrophages; PAS stain: periodic acid-Schiff stain.

**Table 1 cells-10-03044-t001:** Primers of various genes.

Gene Name	Forwards	Reverse
TNFα	5′-GCT CAC AAT GTC TGT GCT TAGAG-3′	5′-GCA GTA GCC ACA GCT CCAG-3′
MCP-1	5′-ATG CAG TTA ATG CCC CAC TC-3′	5′-TGC TGC TGG TGA TTG TCT TG-3′
IL-4	5′-GGA TGT GCC AAA CGT CCT C-3′	5′-GAG TTC TTC TTC AAG CAT GGAG-3′
IL-13	5′-CTT TCT TTA GCG GCC AC-3	5′-CAG AGC GCC ATG AAG CCC AGAG-3′
18S	5′-ACGGAAGGGCACCACCAGGA-3′	5′-CACCACCACCCACGGAATCG-3

TNFα: tumor necrosis factor α (TNFα); MCP-1: monocyte chemoattractant protein-1; IL-4: interleukin-4; IL-13: interleukin-13.

**Table 2 cells-10-03044-t002:** Hemodynamic parameters of cirrhotic rats receiving chronic pioglitazone treatments after acute LPS infusion.

	Sham (n = 4)	Sham+LPS (n = 4)	Sham-Pio+LPS (n = 4)	BDL (n = 9)	BDL+LPS (n = 9)	BDL-Pio+LPS (n = 9)
MAP (mmHg)	110 ± 14	106 ± 9	109 ± 11	92 ± 12 *	87 ± 5 ^#^	90 ± 6
Cardiac output (CO, mL/min)	229 ± 38	203 ± 28	211 ± 19	259 ± 41 *	212 ± 29 ^#^	242 ± 33
PVP (mmHg)	7.8 ± 0.9	9.1 ± 0.6	8.2 ± 0.8	17.3 ± 1.6 *	18.2 ± 2.3	17.9 ± 2.1
RABF (mL/min·100 g)	5.1 ± 1.4	5.6 ± 1.6	5.4 ± 1.2	3.0 ± 1.3 *	3.8 ± 0.5	3.6 ± 0.9
Body weight (BW, gram)	354.6 ± 18.4	349.8 ± 16.5	347.2 ± 13.9	306.4 ± 21.4 *	309.3 ± 18.9	310.2 ± 14.3
Kidney weight (KW, both sides, grams)	1.25 ± 0.007	1.22 ± 14.8	1.21 ± 0.018	1.9 ± 0.078	2.1 ± 0.065	1.4 ± 0.009
KW/BW (10^−3^)	0.36 ± 0.0021	0.35 ± 0.003	0.35 ± 0.007	0.62 ± 0.004 *	0.68 ± 0.003	0.45 ± 0.002 ^#^
Renal hydroxyproline (µg/mg kidney)	312 ± 22	309 ± 27	310 ± 19	429 ± 12 *	418 ± 17	420 ± 8

In sham-LPS or BDL-LPS groups, all measurements were undergone 3 h after LPS infusion; * *p* < 0.05 vs. sham group; ^#^
*p* < 0.05 vs. BDL group. MAP: mean arterial pressure; portal venous pressure (PVP); RABF: renal artery blood flow is the summation of right and left side kidney.

**Table 3 cells-10-03044-t003:** Clinical and serum biochemical data of rats with biliary cirrhosis receiving pioglitazone or vehicle.

	Sham (*n* = 4)	Sham+LPS (*n* = 4) (Mean % Increase from Data of Sham Group)	Sham-Pio+LPS (*n* = 4) (Mean % Increase from Data of Sham Group)	BDL (*n* = 9)	BDL+LPS (*n* = 9) (Mean % Increase from Data of Sham Group)	BDL-Pio+LPS (*n* = 9) (Mean % Increase from Data of Sham Group)
(Endotoxin) (pg/mL)	7.3 ± 0.9	9.6 ± 0.8	8.3 ± 0.5	17.9 ± 2.6 *	27.3 ± 2.8 ^##^	19.4 ± 1.8 ^‡^
(TNFα) (pg/mL)	12.9 ± 5.4	34.8 ± 4.8	17.1 ± 2.8	56.9 ± 8.1 *	168.3 ± 9.6 ^##^	64.1 ± 7.1 ^‡^
(IL-6) (pg/mL)	10.5 ± 1.1	17.6 ± 2.1	14.2 ± 1.6	29.8 ± 3.4 *	73.6 ± 1.9 ^##^	31.8 ± 2.2 ^‡‡^
Fasting blood sugar (mg/dL)	95 ± 15	108 ± 20	98 ± 16	112 ± 23	123 ± 19	119 ± 16
(Albumin) (g/L)	4.1 ± 0.7	3.7 ± 0.9	3.9 ± 0.8	2.9 ± 0.9	2.6 ± 0.4	2.8 ± 0.7
(ALT) (IU/L)	58 ± 14	69 ± 13	61 ± 12	98 ± 7 *	352 ± 15 ^##^	168 ± 12 ^‡‡^
(Total bilirubin) (mg/dL)	0.38 ± 0.09	0.58 ± 0.04	0.41 ± 0.06	7.8 ± 0.8 *	18.5 ± 1.7 ^##^	13.1 ± 2.5 ^‡^

In sham-LPS or BDL-LPS groups, blood was collected for various measurement 3 h after LPS infusion; * *p* < 0.05 vs. sham group; ^##^
*p* < 0.001 vs. BDL group; ^‡^, ^‡‡^
*p* < 0.05, 0.001 vs. BDL+LPS group; TNFα: tumor necrosis factor α (TNFα); IL-6: interleukin-6; ALT: alanine aminotransferase.

## Data Availability

The datasets in this study are available from the corresponding author upon reasonable request.

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
