# Peer review of "Pioglitazone Ameliorates Acute Endotoxemia-Induced Acute on Chronic Renal Dysfunction in Cirrhotic Ascitic Rats"

_cells, 2021, doi:10.3390/cells10113044_

Round 1
Reviewer 1 Report
This manuscript study the role of pioglitazone in the normalization of renal arterial levels of PPARγ and consequently has a preventive effect on LPS-induced acute and chronic renal dysfunction in BDL-induced cirrhosis model in rats. The manuscript is well written and obtained results are interesting. However, several points need to be clarified in order to improve the readability of the manuscript. These points are:
a) there are several incorrect words such as "hepatopulmlonary" on page 2. Please review possible mistakes throughout the text
b) data are described as mean and sem and differences between groups were compared using Student t test. Considering the reduced number of rats by group and that many of experimental studied variables do not fit to a normal distribution may be wrong to describe data as means and sem. Also, non-parametric test should be used instead t test if variables do not fit to normal distribution
c) data comparisons among multiple groups such as is shown on tables or figures using t test is a statistic problem (multiple comparisons). Authors should use a statistical test able to control this problem such as ANOVA with post-hoc test (if residuals fit a normal distribution else a non-parametric test should be used)
d) Do the authors analyzed any information about changes and injury in liver and their correlation with kidney changes?. A brief discussion about this possible association must be included.
Author Response
Ref.: Ms. No. Manuscript ID: cells-1453943R1
Reviewer 1: Comments and Suggestions for Authors
This manuscript study the role of pioglitazone in the normalization of renal arterial levels of PPARγ and consequently has a preventive effect on LPS-induced acute and chronic renal dysfunction in BDL-induced cirrhosis model in rats. The manuscript is well written and obtained results are interesting. However, several points need to be clarified in order to improve the readability of the manuscript. These points are:
Response: Thanks for giving us this opportunity to revise our manuscript. All your instructions improve our manuscript. The point-to-point response are included as below.
Comment a): there are several incorrect words such as "hepatopulmlonary" on page 2. Please review possible mistakes throughout the text.
Answer to comment a): Thanks for your very useful comment. In “revised” version, we had corrected the “hepatopulmonary” on page 2. Meanwhile, the whole text had been checked carefully. The wording and grammar were checked and corrected again.
Comment b): data are described as mean and sem and differences between groups were compared using Student t test. Considering the reduced number of rats by group and that many of experimental studied variables do not fit to a normal distribution may be wrong to describe data as means and sem. Also, non-parametric test should be used instead t test if variables do not fit to normal distribution
Answer to comment b): Thanks for your very useful comments. According to your suggestion, we had used the Mann-Whitney U test for the comparison of the data of mean/SEM and differences two groups and ANOVA with post-hoc test for comparison among multiple groups. The description and results were included in manuscript (line 175-177).
Comment c) data comparisons among multiple groups such as is shown on tables or figures using t test is a statistic problem (multiple comparisons). Authors should use a statistical test able to control this problem such as ANOVA with post-hoc test (if residuals fit a normal distribution else a non-parametric test should be used)
Answer to comment c): Thanks for your very useful comments. According to your suggestion, we had used the Mann-Whitney U test for the comparison of the data of mean/SEM between two groups and ANOVA with post-hoc test for comparison among multiple groups in Tables and Figures.
Comment d) Do the authors analyzed any information about changes and injury in liver and their correlation with kidney changes?. A brief discussion about this possible association must be included.
Answer to comment d): Thanks for giving us this opportunity to discuss this important issue. As shown in Table 2, in comparison with BDL group, the acute intraperitoneal (IP) infusion of LPS induced the mean 195% (56.9±8.1 vs. 168.3±9.6 pg/mL) increase in serum TNFα level in BDL+LPS group whereas LPS only induced 12% (56.9±8.1 vs. 64.1±7.1 pg/mL) increase in serum TNFα level in BDL-pio+LPS group. This result indicated that pioglitazone treatment prevented the 183% of LPS-induced elevation of serum TNFα level in cirrhotic rats. Additionally, in comparison with liver injury marker (serum ALT level) of BDL group, the 259% of elevation of liver injury marker in BDL+LPS group was suppressed to 71% which indicated that the pioglitazone treatment prevented the 186% of LPS-induced elevation of serum ALT (Table 2). Similarly, the pioglitazone treatment suppressed the 40% of LPS-elevated renal injury marker (renal IL-18 expression) in BDL-pio+LPS group. Particularly, higher degree of pioglitazone-suppressed liver injury marker (serum ALT) than renal marker (IL-18) might be resulted from direct access of LPS to the portal system by the IP administration [Shoyaib, A.A. , Archie, S.R., Karamyan, V.T. Intraperitoneal route of drug administration: should it be used in experimental animal studies? Pharm. Res. 2019, 37(1), 12]. These results indicated that the circulating TNFα mediated the LPS-induced liver and renal injuries and these changes of cirrhotic rats in this study can be ameliorated by pioglitazone treatment. Above description had been included in line 318-332.

Reviewer 2 Report
Liu et al. presented a study on the beneficial effects of chronic pioglitazone administration on kidney injury in cirrhotic rats. The study is well conducted and nicely presented, giving new evidences which could be useful for future studies. There are some minor suggestion that I would like to give to the authors to improve the discussion:
- In the discussion section it would be interesting to have a brief speculation of the authors on the possible implications of these results on future studies. For exemple, the use of pioglitazone in non alcoholic steatohepatitis should be briefly discussed to support further studies on the beneficial effects of this drug, which is still not approved for clinical use in many countries
- Pioglitazone was administered iv. The authors should comment on whether they think these results could be reproduced by oral administration of this drug.
- MINOR: the abbreviation SEM page 4 line 171 should be defined
- Abbreviations should be defined in the legenda of each table and figure to facilitate their interpretation to readers
Author Response
Ref.: Ms. No. Manuscript ID: cells-1453943R1
Reviewer 2: Comments and Suggestions for Authors
Liu et al. presented a study on the beneficial effects of chronic pioglitazone administration on kidney injury in cirrhotic rats. The study is well conducted and nicely presented, giving new evidences which could be useful for future studies. There are some minor suggestion that I would like to give to the authors to improve the discussion:
Response: Thanks for giving us this opportunity to revise our manuscript. All your instructions improve our manuscript. The point-to-point response are included as below.
Comment 1: In the discussion section it would be interesting to have a brief speculation of the authors on the possible implications of these results on future studies. For example, the use of pioglitazone in non-alcoholic steatohepatitis should be briefly discussed to support further studies on the beneficial effects of this drug, which is still not approved for clinical use in many countries
Response 1: Thanks for giving us this opportunity to include the description of in discussion section. We had been included these descriptions in the “revised” version. The discussion as below “High prevalence of renal dysfunction had been reported among non-alcoholic steatohepatitis (NASH) patients [Metabolism 2018, 79, 64-76]. The severe NASH is the most rapidly growing indication for simultaneous liver kidney transplantation with poor renal outcomes [J. Hepatol. 2020, 72(4), 785-801]. Several large-scale randomized controlled trials have reported the effectiveness of pioglitazone in treating NASH to improve markers of hepatic steatosis and fibrosis on liver histology [BMJ. Open Diabetes Res. Care 2021, 9(1), e001990; Transplantation 2016, 100, 607-612]. Recent study reported that low dose of pioglitazone was safe and effective in diabetes patients with CKD [PLoS One 2018, 13(10), e0206722]. Especially, pioglitazone decreases the incidence of newly onset end-stage renal disease in diabetes patients [Ann. Intern. Med. 2016, 165, 305–315]. Renal dysfunction in NASH and diabetic patients shared common pathogenesis [Diabetes Care 2020, 43(10), e152-e153; J. Clin. Exp. Hepatol. 2019, 9(1), 22–28]. Taken together, the pioglitazone might have the potentials to protect NASH patients from the development of renal dysfunction which need to be evaluated in future studies” had been included in “revised” manuscript. The additional references 51-57 had been included “revised” version.
Comment 2: Pioglitazone was administered iv. The authors should comment on whether they think these results could be reproduced by oral administration of this drug.
Response 2: Thanks for your very important comment for the route of drug administration in our study. In this study, 2 weeks of pioglitazone (MW 392.9) was administered by intraperitoneal (IP) with azert osmotic pump. Pioglitazone is well absorbed, has an oral bioavailability of about 80%, and is extensively metabolized to active and inactive metabolites in the liver [Eckland, D. A., and Danhof, M. (2000). Clinical pharmacokinetics of pioglitazone. Exp. Clin. Endocrinol. Diabetes 108, S234–S242; Hanefeld M. Pharmacokinetics and clinical efficacy of pioglitazone. Int J Clin Pract Suppl 2001;(121):19-2; Yoneda M, et al. Comparing the effects of tofogliflozin and pioglitazone in non-alcoholic fatty liver disease patients with type 2 diabetes mellitus (ToPiND study): a randomized prospective open-label controlled trial. BMJ. Open Diabetes Res. Care 2021, 9(1), e001990; Satirapoj B, Watanakijthavonkul K, Supasyndh O. Safety and efficacy of low dose pioglitazone compared with standard dose pioglitazone in type 2 diabetes with chronic kidney disease: A randomized controlled trial. PLoS One 2018, 13(10), e0206722]. In most cases, IP administration of drugs in experimental animals is a justifiable route for pharmacological and proof-of-concept studies where the goal is to evaluate the effect(s) of target engagement rather than properties of a drug formulation and/or its pharmacokinetics for clinical translation. Previous study had reported that the higher bioavailability and absorption of IP route of small molecular agent (MW <5000) are higher than those by oral route. But both IP and oral routes have similar first pass metabolism of small molecular agents in the liver [Shoyaib AA , Archie SR, Karamyan VT. Intraperitoneal Route of Drug Administration: Should it Be Used in Experimental Animal Studies? Pharm Res 2019;37(1):12]. In comparison with oral route, IP technique is easy to master and minimally stressful for animals. IP route is especially commonly used in chronic studies involving rats for which repetitive oral access is challenging. We agreed with your opinion that the effectiveness of IP and oral administration of 2 weeks of pioglitazone need to be compared in future studies. Above discussion had been included in “revised” manuscript [line 364-378].
Comment MINOR: the abbreviation SEM page 4 line 171 should be defined.
Response: Thanks for your reminding about the abbreviation SEM. We had included the full name of SEM in “revised” version.
Comment : Abbreviations should be defined in the legenda of each table and figure to facilitate their interpretation to readers
Response: According to your instruction, all the abbreviations in the legenda of each table and figure had been spelled out to facilitate their interpretation to readers
